# Differences in Glycolysis and Mitochondrial Respiration between Cytotrophoblast and Syncytiotrophoblast In-Vitro: Evidence for Sexual Dimorphism

**DOI:** 10.3390/ijms221910875

**Published:** 2021-10-08

**Authors:** Matthew Bucher, Leena Kadam, Kylia Ahuna, Leslie Myatt

**Affiliations:** Department of Obstetrics and Gynecology, Oregon Health and Science University, Portland, OR 97239, USA; mbucher@uoregon.edu (M.B.); kadam@ohsu.edu (L.K.); ahuna@ohsu.edu (K.A.)

**Keywords:** placenta, metabolism, glycolysis, mitochondrial respiration, cytotrophoblast, syncytiotrophoblast, placental metabolism, trophoblast glycolysis, trophoblast mitochondrial respiration, sexual dimorphism

## Abstract

In the placenta the proliferative cytotrophoblast cells fuse into the terminally differentiated syncytiotrophoblast layer which undertakes several energy-intensive functions including nutrient uptake and transfer and hormone synthesis. We used Seahorse glycolytic and mitochondrial stress tests on trophoblast cells isolated at term from women of healthy weight to evaluate if cytotrophoblast (CT) and syncytiotrophoblast (ST) have different bioenergetic strategies, given their different functions. Whereas there are no differences in basal glycolysis, CT have significantly greater glycolytic capacity and reserve than ST. In contrast, ST have significantly higher basal, ATP-coupled and maximal mitochondrial respiration and spare capacity than CT. Consequently, under stress conditions CT can increase energy generation via its higher glycolytic capacity whereas ST can use its higher and more efficient mitochondrial respiration capacity. We have previously shown that with adverse in utero conditions of diabetes and obesity trophoblast respiration is sexually dimorphic. We found no differences in glycolytic parameters between sexes and no difference in mitochondrial respiration parameters other than increases seen upon syncytialization appear to be greater in females. There were differences in metabolic flexibility, i.e., the ability to use glucose, glutamine, or fatty acids, seen upon syncytialization between the sexes with increased flexibility in female trophoblast suggesting a better ability to adapt to changes in nutrient supply.

## 1. Introduction

The placenta is a highly specialized fetal organ responsible for supporting growth and development of the fetus in utero. It forms an immune and physical barrier between the mother and fetus and provides metabolic, transport, and endocrine functions [1]. Each of these functions exact a metabolic cost so it comes with no surprise that the placenta has an extraordinarily high metabolic rate, accounting for approximately 40% of the total oxygen consumed by the fetus and placenta combined [2]. The chorionic villi of the placenta, which contain fetal capillaries, are bathed with maternal blood and it is here on its outer surface where oxygen and nutrient uptake and transfer between maternal and fetal circulations occurs across the syncytiotrophoblast (ST) cell layer, a multinuclear syncytium 13 m^2^ in area at term. The ST also synthesizes and secretes large amounts of peptide and steroid hormones [3]. The ST cell layer is formed from underlying mononucleated villous cytotrophoblast (CT) cells which constantly proliferate in vivo and fuse into the fully differentiated multinucleated ST. Formation of ST can be recapitulated in vitro by culture of isolated CT cells which, although they cannot proliferate in-vitro, spontaneously fuse, and differentiate into ST [4]. As the majority of energy requiring placental functions (nutrient/waste transfer and hormone production) are carried out by the ST layer, it has traditionally been assumed that the ST is more metabolically active than CT cells, although recent studies suggest a more complex metabolic status [5].

Trophoblast cells, like other cells, produce chemical energy in the form of adenosine triphosphate (ATP) mainly via oxidative phosphorylation. Glycolysis, the TCA cycle, and fatty acid oxidation all result in formation of energy-rich NADH and FADH2 which donate their electrons into the electron transport chain (ETC) for shuttling down a chain of protein complexes while protons are pumped out of the mitochondrial matrix into the intermembrane space, creating a proton gradient across the inner mitochondrial membrane. In the final step of oxidative phosphorylation, protons travel down their concentration gradient through complex 5 (ATP synthase) and phosphorylate adenosine diphosphate (ADP), creating ATP. In most cells, breakdown of glucose via glycolysis and formation of acetyl CoA is the primary pathway that provides metabolites for oxidative phosphorylation. However, cells can switch to other metabolites in either the absence of glucose or excess of fatty acids. This ability to switch metabolite substrates depending on nutrient availability is called metabolic flexibility and is a crucial cell survival mechanism when faced with sub-optimal metabolic conditions. We recently showed that in addition to glucose, trophoblast cells can also utilize amino acids, e.g., glutamine, and fatty acids for generation of ATP via the ETC and that the proportions of each used can change with metabolic condition, e.g., obesity or gestational diabetes [6]. Since, the proliferative CT and differentiated ST have different role in terms of transport, metabolism, and steroid and peptide hormone production, we hypothesized that they might differ in their use of fuel sources and metabolic flexibility.

Cytotrophoblast cells share many similarities with cancer cells which proliferate, migrate, and invade tissues to establish a continuous nutrient supply to support the development of a tumor. In-vivo, CT proliferate, migrate, and invade (as extravillous trophoblast) endometrial tissue to establish a nutrient supply but also as villous cytotrophoblast undergo fusion to form ST [7]. Otto Warburg described a phenomenon, the Warburg effect, where cancer cells preferentially utilize glycolysis in the presence of oxygen (aerobic glycolysis) to produce the bulk of their ATP requirement, unlike normal body cells that generate ATP through mitochondrial respiration using metabolites from glycolysis, the TCA cycle and β-oxidation of fatty acids [8,9,10]. Based on the similarities between CT cells and cancer cells, we therefore hypothesized that CT might have higher glycolytic function, compared to ST cells.

There is now an overwhelming body of data indicating a sexual dimorphism exists in placental physiology underpinned by a sex-dependent difference in placental gene expression [11,12,13,14]. This may be linked to the different fetal growth and survival strategies where male fetuses grow larger than female fetuses but are therefore at a greater risk of suffering from adverse pregnancy outcomes if maternal nutrition and placental function are not optimal [15,16,17]. We have previously reported maternal obesity, preeclampsia, and gestational diabetes mellitus to be associated with sexually dimorphic effects on energetics and autophagy in the placenta, and have also shown sexual dimorphism in placental antioxidant enzyme activity [6,18,19,20]. In this study we also investigated if fetal sex had effects on glycolytic and mitochondrial metabolism in CT vs. ST cells from women of a healthy weight.

## 2. Results

### 2.1. Clinical Characteristics

All the women who donated their placentas to this study were chosen because they were of healthy pre-pregnancy (lean) BMI (<25 kg/m^2^). There were no significant differences in gestational age at delivery, maternal age, pre-pregnancy BMI (Body Mass Index), gestational weight gain, or placental weight between the groups (Table 1). However, there were significant differences in fetal weight and the fetal/placental weight ratio between male vs. female pregnancies with the male being significantly heavier and, hence, with a more “efficient” placenta. 

On average, male fetuses are born larger than female fetuses [21], with little differences in placental weight, resulting in a larger fetal to placental weight ratio in males [22]. Our data agrees with these findings (Table 1).

### 2.2. Isolated Cytotrophoblast Differentiate into Syncytiotrophoblast in Culture

Isolating intact ST from the placenta is not feasible as the digestion process destroys the syncytial layer. However, CT can be isolated and in culture will aggregate and fuse to form ST over 96 hrs. Figure 1A shows individual cells positive for cytokeratin-7 confirming identity as single CT at 24 hrs. Over the course of the culture, these undergo fusion to form ST as evidenced by multinucleate structures with positive cytokeratin-7 stain (Figure 1B,C) and E-cadherin stain (Appendix A).

To further verify that our technique of culturing trophoblasts results in ST formation, we measured human chorionic gonadotropin (hCG) production. With data from both fetal sexes combined, ST, as expected had significantly higher hCG production (*p* = 0.007) compared to CT (Figure 2D). With fetal sex separated, ST from both males (*p* = 0.01) and females (*p* = 0.02) have significantly increased hCG production, compared to CT of the same sex (Appendix A) however interestingly, the increase in hCG production upon syncytialization appears to be greater in female vs. male trophoblast (*p* = 0.02).

### 2.3. Cytotrophoblast Have Higher Glycolytic Capacity and Reserve Capacity

The glycolytic function of CT and ST cells was measured using the glycolysis stress test (Figure 2A). When analyzing with fetal sex combined, no differences were observed in non-glycolytic acidification or rates of glycolysis (Figure 2B,C) suggesting both CT and ST have similar rates of basal glycolysis and basal bioenergetics. However, CT showed significantly higher glycolytic capacity (*p* = 0.01) and glycolytic reserve (*p* = 0.0003) when compared to ST (Figure 2D,E, Appendix A). Glycolytic capacity indicates the maximum amount of glycolysis/glucose breakdown the cells can perform acutely, whereas glycolytic reserve (glycolytic capacity−glycolysis rate) is the difference between the basal and maximal glycolytic capacity. The glycolytic reserve thus indicates the cells potential to increase ATP production via glycolysis under stress or other physiologically energy-demanding situations. Our results thus suggest that whereas CT and ST have similar basal rates of glycolysis, CT have higher potential for energy/ATP generation via glycolysis when stressed. 

We then separated the data to determine the effects of fetal sex (Appendix A). Non-glycolytic acidification and basal glycolysis rate which were not different between CT and ST were also not different between the sexes (Appendix A). Male CT however showed significantly higher glycolytic capacity (*p* = 0.04) when compared to their ST whereas no difference was observed between the female CT and ST. Interestingly, there was no sexually dimorphic effect on glycolytic reserve as male (*p* = 0.015) and female ST (*p* = 0.039) both had significantly lower reserve as compared to their CT, suggesting that under energetically demanding or stressed conditions, both male and female ST have less potential to use glycolysis for ATP production (Appendix A). 

### 2.4. Syncytiotrophoblast Have Higher Mitochondrial Respiration Compared to Cytotrophoblast

The Mitochondrial stress assay was performed to determine how mitochondrial oxidative respiration and the resultant ATP production change as CT differentiate to ST (Figure 3A). With data from both fetal sexes combined, ST had significantly higher basal respiration (oxygen consumption in the resting state) (*p* = 0.003) and higher ATP-coupled respiration (*p* = 0.0008), suggesting ST are energetically more demanding than CT (Figure 3B,C, Appendix A). In addition, the ST also showed significantly higher maximal respiration (*p* = 0.0001) and spare capacity (*p* = 0.0001), suggesting that ST can achieve a higher rate of mitochondrial respiration if needed and have a higher ability to respond to demand when compared to CT (Figure 3D,E). Syncytiotrophoblast also showed significantly higher non-mitochondrial respiration (*p* = 0.009) and proton leak (*p* = 0.04), compared to CT (Figure 3F,G). Proton leak is the amount of oxygen consumption not coupled to ATP production in the mitochondria and has been linked to the levels of reactive oxygen species (ROS) and oxidative stress in the cell [23,24,25]. 

To determine the effect fetal sex has on mitochondrial function, data were analyzed separately for male and female groups (Appendix A). Overall, ST from both males and females showed trends similar to that observed in the combined analysis. In both sexes, ST had significantly higher ATP-coupled respiration (M, *p* = 0.03, F, and *p* = 0.01), maximal respiration (M, *p* = 0.007, F, and *p* = 0.007) and spare capacity (M, *p* = 0.016, F, and *p* = 0.007) compared to CT. In females, ST had significantly higher basal respiration (*p* = 0.02) and non-mitochondrial respiration (*p* = 0.03) compared to CT. In males, ST had significantly higher proton leak (*p* = 0.03) compared to CT (Appendix A). 

### 2.5. Cytotrophoblast and Syncytiotrophoblast Differ in Their Capacity to Respond to Stress

To more clearly visualize how the metabolic phenotype changes as CT fuse to form ST, basal OCR vs. basal ECAR measurements were plotted against each other (Figure 4A). Both male and female trophoblasts increase glycolysis (ECAR) and oxidative phosphorylation (OCR) upon syncytialization showing the increased energy demands upon fusion into ST. The metabolic potential of CT (Figure 4B) and ST (Figure 4C) in response to stress was visualized by plotting basal OCR/ECAR rates and maximal OCR/ ECAR rates upon addition of FCCP or Oligomycin, respectively, mimicking a physiologically stressed situation. CT did not show any increase in their oxygen consumption rate (OCR), a direct measure of their ability to generate ATP via oxidative phosphorylation, when stressed by addition of FCCP but did show increased glycolytic function (ECAR, M, *p* = 0.02, F, and *p* = 0.008). This is expected as CT do not have appreciable mitochondrial spare capacity (Figure 3E) but do have appreciable glycolytic reserve (Figure 2E). In contrast, ST (Figure 4C) can increase oxidative phosphorylation (OCR) as well as glycolysis (ECAR) to meet energy demands when presented with a stress (M, *p* = 0.11, F, and *p* = 0.05). This is also expected as ST have significantly higher mitochondrial spare capacity (Figure 3E) and some glycolytic reserve, although less glycolytic reserve than CT (Figure 2E). Taken together, these results suggest that CT and ST have different strategies and capabilities to respond to physiologically demanding/stressful conditions. 

### 2.6. Syncytiotrophoblast Have a Higher Capacity and Efficiency for Substrate Utilization under Stress

We examined the impact of different substrates on trophoblast spare respiratory capacity (the ability to respond to increased energy demand or under stress) using an adapted version of the mitochondrial stress test. A combination of two pathway inhibitors (UK5099 (glucose), BPTES (glutamine) or Etomoxir (long-chain fatty acid)) was added, leaving only a single substrate utilization pathway open, before performing the mitochondrial stress test protocol. The impact of availability of just a single substrate—glucose or glutamine or fatty acids—on the ability of cells to increase respiration rate in physiologically stressed conditions (spare/reserve capacity) was then calculated as described in the methods section.

With both sexes combined, CT had a significantly higher capacity to use either glucose (*p* = 0.01) or fatty acids (*p* = 0.04) alone than glutamine alone. However ST were observed to have a significantly higher spare capacity for glucose utilization than glutamine (*p* < 0.02) or fatty acids (*p* < 0.05) when each was present alone which indicates a higher efficiency in glucose utilization over the other two substrates. When compared to CT, ST also had significantly higher spare capacity when glucose (*p* = 0.02) or glutamine (*p* = 0.003) were available alone as substrates (Figure 5A) with no difference apparent between CT and ST spare capacity when only fatty acids were available as substrate. These results suggest that upon syncytialization there is a change in the preference for utilization of the different substrates to generate energy under physiologically stressed conditions. 

We then separated the data based on fetal sex to determine if the cells differed in their capacities for substrate utilization under stress (Figure 5B,C). In males, ST showed a trend towards an increase in spare capacity for each substrate over CT, but the values reached significance only for glutamine (*p* = 0.04) (Figure 5B). Interestingly, in females CT had a higher spare capacity (and hence efficiency in utilization) for glucose (*p* = 0.03) over glutamine and a trend towards increased spare capacity for long-chain fatty acids over glutamine (*p* = 0.09). Female ST had significantly higher capacity for use of glucose compared to glutamine (*p* = 0.01) and long-chain fatty acids (*p* = 0.01) (Figure 5C). Thus, it appears that female trophoblast account for most changes in substrate utilization under stress conditions when male and female are combined.

### 2.7. Syncytiotrophoblast Have Lower Mitochondrial Content but Higher Citrate Synthase Activity

To determine if the increased overall mitochondrial respiration observed in ST was a function of increased number of mitochondria, we measured mitochondrial content using the mitochondria specific dye MitoTracker^TM^ (normalized to total DNA amount). Surprisingly, we found the opposite to be true. With data from both fetal sexes combined, CT have significantly greater mitochondrial content compared to ST (*p* = 0.007) (Figure 6A). However, when separated by fetal sex, CT from males (*p* = 0.01) account for the majority of this difference with significantly higher mitochondrial content compared to ST, while females only approached significance (*p* = 0.07) (Appendix A).

To further validate the above observation, we quantified the expression using western blotting of two other mitochondrial markers, citrate synthase, and voltage-dependent anion channel (VDAC) found in the mitochondrial outer membrane. In agreement with the MitoTracker^TM^ data, the ST had lower expression of both citrate synthase (*p* = 0.01) and VDAC (*p* = 0.007) (Figure 6B,C). When the data was separated and analyzed based on fetal sex the decrease in citrate synthase expression upon syncytialization was significant only in male mirroring the change seen with MitoTracker^TM^ whereas VDAC significantly decreased in both male and female trophoblast with syncytialization (Appendix A). 

We subsequently measured citrate synthase activity as an additional marker for overall mitochondrial activity. Citrate synthase is responsible for catalyzing the first step of the citric acid cycle by combining acetyl-CoA (end product of all three fuel oxidation pathways) with oxaloacetate to generate citrate which then enters the TCA cycle to generate FADH2 and NADH. With data from both sexes combined, ST have significantly higher citrate synthase activity (*p* = 0.007) compared to CT (Figure 6D), however, separation by fetal sex revealed male (*p* = 0.008) ST have significantly increased citrate synthase activity compared to CT, while female ST only approached significance (*p* = 0.09) (Appendix A). Increased citrate synthase activity in ST aligns with our results of increased mitochondrial respiration rate in ST.

### 2.8. Effect of Syncytialization on Mitochondrial Protein Expression

We next investigated if the increased mitochondrial respiration and citrate synthase activity measured in ST corresponded with an increase in the expression of proteins involved in mitochondrial catabolic pathways (outlined in Table 2).

Surprisingly, we also found that every mitochondrial specific protein we measured significantly decreased in ST compared to CT. As seen in Figure 7, the expression of all five complexes in the respiratory chain, I. NADH dehydrogenase (*p* = 0.007), II. Succinate dehydrogenase (*p* = 0.007), III. Cytochrome C reductase (*p* = 0.02), IV. Cytochrome C oxidase (*p* = 0.007) and V. ATP synthase (*p* = 0.01) significantly decrease in ST compared to CT (Figure 7E–I). Glutaminase and glutamate dehydrogenases (GLUD 1/2) the mitochondrial enzymes that convert glutamine into glutamate, and glutamate to α-ketoglutarate, a precursor of the citric acid cycle intermediate, respectively, were also found to be significantly decreased in ST compared to CT (*p* = 0.0078,) (Figure 7J,K). However, no differences were observed in Hexokinase 2, Carnitine palmitoyltransferase one alpha (CPT1α), or Glucose Transporter Type 1 (GLUT1) expression (Figure 7L–N). Peroxisome proliferator-activated receptor gamma coactivator 1-alpha (PGC1α), which is the master regulator of mitochondrial biogenesis, was also found to be significantly decreased in ST compared to CT (*p* = 0.007) (Figure 7O). 

Similar observations were made when data was separated by fetal sex (Appendix A). Both male and female ST had significantly decreased protein expression of NADH dehydrogenase (M, *p* = 0.006, F, and *p* = 0.001), Succinate dehydrogenase (M, *p* = 0.003, F, and *p* = 0.001), Cytochrome C oxidase (M, *p* = 0.01, F, and *p* = 0.001), GLUD1/2 (M, *p* = 0.01, F, and *p* = 0.008), Glutaminase (M, *p* = 0.002, F, and *p* = 0.02) and PGC1α (M, *p* = 0.03, F, and *p* = 0.0005) compared to CT of the same fetal sex. Male ST had significantly decreased Glucose transporter 1 (*p* = 0.029) while female ST had significantly decreased ATP synthase (*p* = 0.02) and trended to have decreased Cytochrome C reductase (*p* = 0.09). No differences were seen in CPT1α or Hexokinase 2 across any of the groups.

## 3. Discussion

Cell differentiation and differentiated functions are highly energy-consuming processes [26]. Several studies have reported significant modifications in cellular bioenergetics as progenitor cells differentiate [27,28]. However, the shifts in mitochondrial and cellular bioenergetic pathways during ST differentiation are not well understood. Additionally, while sexual dimorphism in placental function has been reported, the effect of fetal sex on CT and ST bioenergetics and mitochondrial function has been largely unexplored. 

The present study provides several lines of evidence that CT and ST differ significantly in their metabolic phenotypes. CT have equivalent basal glycolysis but a higher glycolytic capacity and reserve than ST whereas ST have a higher mitochondrial respiratory function than CT under both basal conditions and conditions mimicking physiological stress and increased energy demand. ST also appear to utilize glucose and glutamine more efficiently than CT whereas the two cell types show no difference in their ability to use fatty acids to generate energy. Further, both CT and ST show a distinct sexual dimorphism in their energy metabolism with male ST having lower glycolytic capacity and reserve compared to their CT and with female ST having comparable glycolytic capacity, but lower reserve than their CT. On the other hand, both male and female ST have higher mitochondrial respiration (compared to their respective CT) for all parameters except basal respiration which is not different in male ST vs. CT and proton leak which is not different in female ST vs. CT. 

In the current study, we used isolated term CT cells cultured for 24 h and 96 h representing progenitor CT cells and syncytialized ST, respectively. Syncytialization over this timeframe was confirmed by staining for the trophoblast marker CK-7 and for nuclear aggregates and measuring hCG secretion as shown in Figure 1. We then assessed glycolytic function and mitochondrial respiration in both CT and ST using the Seahorse assay. The assay measures the rate of depletion of O_2_ from the media, “oxygen consumption rate” (OCR) and protons released into the media, “extracellular acidification rate” (ECAR) as indicators of mitochondrial oxidative phosphorylation and glycolytic function, respectively. 

Although, there was no statistical difference in the basal rate of glycolysis between CT and ST, we observed that CT had a significantly higher glycolytic capacity and reserve capacity than ST (Figure 2). Kolahi et al. previously reported significantly higher basal glycolysis rate in CT but no difference in the glycolytic reserve. However, their study was performed with media containing pyruvate, a product in the glycolysis pathway which upon breakdown releases lactate and proton measured as ECAR in the Seahorse assay. The presence of pyruvate would thus affect the baseline measurements performed in the study and may account for the differences seen in this study. Higher glycolytic capacity and reserve in CT suggests that under physiologically energy demanding conditions, CT but not ST could rapidly increase their glycolytic function to survive. From a bioenergetic perspective, glycolysis is not as efficient as mitochondrial respiration for ATP production with 2 vs. 36 ATP molecules being generated per glucose molecule respectively. However, it is generally accepted that proliferating progenitor cells, such as cytotrophoblast, are glycolytic in nature [29,30,31,32] and it is the preferred way to generate ATP in cancer cells described as the Warburg effect [8]. 

We also observed that the differentiated ST have significantly higher levels of both basal mitochondrial respiration and higher reserve capacity (Figure 3). ST were also more flexible in their fuel dependency and were able to better utilize glucose and glutamine for energy generation under conditions mimicking physiological stress and energy demand (Figure 5). Studies assessing bioenergetics of neuronal, osteogenic, and erythroid differentiation also made similar observations where differentiation was accompanied by, and required, a shift towards mitochondrial respiration [27,28,32,33]. Interestingly, previous studies on mitochondrial function in human placenta have reported observations contradictory to ours. Fisher et al. reported reduced oxygen consumption, mitochondrial respiration, and ATP production in mitochondria of ST vs. CT [34]. However, these studies use intact mitochondria isolated from whole placental tissue by homogenization followed by density gradient purification. This separates bigger mitochondria from small mitochondria which the authors refer to as cyto–mito and syncytio–mito, respectively, based on previous studies that reported large circular mitochondria in cytotrophoblast and small irregular shaped ones in syncytiotrophoblast [35]. As prepared this “cyto–mito” fraction will also contain normal sized mitochondria found in other cell types of the placenta such as stromal and endothelial cells and hence does not only represent cytotrophoblast mitochondrial respiration. Our use of intact cells (individual or syncytialized) better mimics the physiological milieu which may have impact on availability of substrates, membrane potential and proton gradients all crucial for mitochondrial function. Similarly, Kolahi et al. reported reduced OCR in ST contradictory to our observations. However, their study was focused on fatty acid metabolism and assays contained high concentrations of saturated long chain and monounsaturated fatty acids which could account for the observed differences in OCR seen. 

The differentiated functions of syncytiotrophoblast means their mitochondria have several functions distinct from those of the proliferative cytotrophoblast, particularly steroidogenesis. Martinez et al. reported that ST mitochondria have significantly increased cytochrome P450 expression an enzyme responsible for catalyzing the first step in steroidogenesis, highlighting the role of syncytiotrophoblast in hormone synthesis [35]. Similarly, Fisher et al. reported increased CYP11A1 expression and increased progesterone production in ST mitochondria. While we did not assess steroidogenesis, we measured citrate synthase activity as a marker for mitochondrial activity. Like the above studies, we found that ST had higher citrate synthase activity, again implying greater mitochondrial function. However, our assessment showed significantly reduced mitochondrial content, as well as decreased protein expression of citrate synthase, VDAC, mitochondrial ETC complexes and other enzymes involved in mitochondrial respiration seemingly counter-intuitive to our results on mitochondrial function. (Figure 6 and Figure 7, Appendix A). We assessed mitochondrial content using the specific dye MitoTracker^TM^ deep red and normalizing it to the nuclear DNA content determined by the Hoechst DNA stain. The dye accumulates in active mitochondria and is used for mitochondrial tracking in live cells. Its correlation to mitochondrial mass is, however, not clearly defined. Complementing our results with additional assays such as quantifying cardiolipin content or ratio of mitochondrial DNA to nuclear DNA might provide a better idea of mitochondrial mass in ST [36]. 

Several studies have also highlighted how mitochondrial ultrastructure and cristae organization play a critical role in its function (comprehensive review in [37]). Detailed ultrastructure studies using cryo-electron tomography have suggested that ATP synthase dimers preferentially localize in (and even aid in formation of) the curved regions of the cristae, such as the tips, whereas the ETC complexes are in less curved regions, such as the stalks [38,39,40,41]. These observations suggest that the cristae structure is finely tuned to support the energetic needs of the respective cells. Increased number of cristae could improve mitochondrial function but substantially reduce the available matrix space for metabolic enzymes [37] which would explain the reduced expression but increased function observed in our study. Recently, Cagiliati et al. showed that cristae structure drives the assembly of respiratory chain super complexes (RCS) (consisting of ATP synthase and ETC enzymes) on their surface and therefore affect the efficiency of mitochondrial respiration [42]. They further reported that mitochondrial fusion protein OPA1 (Optic Atrophy Protein 1) was crucial for cristae organization and structure, assembly of the RCS, and respiratory function. Increased expression of another fusion protein mitochondrial fusion protein-2 (Mfn2) has also been correlated to increased mitochondrial function further emphasizing the correlation between mitochondrial ultrastructure, function and ‘mitochondria-shaping’ proteins that regulate the organelle’s fission and fusion [43]. We propose that a detailed analysis of ST and CT mitochondrial cristae structure and studying expression of mitochondrial shaping proteins might provide further insights into the above results. 

An important aim for the study was to assess sexual dimorphism, if any, in placental mitochondrial function. Sexual dimorphism in fetal and placental development as well as placental gene expression has been reported before [14,44]. Male fetuses are known to be bigger and heavier than females with equivalent placental weight as observed in our study [21,22,45] and are therefore considered more efficient, but vulnerable to gestational stressors. Placental responses to environmental stress, such as hyperlipidemia and asthma, are influenced by fetal sex wherein female fetus growth is limited increasing the chances of survival, but male fetuses continue growing normally, increasing their chances of a poor outcome in case of acute exacerbation of the stressors [11,16,46]. We have previously shown that indeed male and female syncytiotrophoblast show differences in metabolic flexibility in use of glucose, glutamine, or fatty acids when they are exposed to different intrauterine environments, i.e., from normal weight, obese, or type A2 gestational diabetes, with male trophoblasts being more severely affected [6,14,47]. To the best of our knowledge, this is the first study assessing sexual dimorphism in basal mitochondrial function and response to stressors as CT from normal pregnancies differentiate to ST. We report that when CT differentiate into ST, they reduce their glycolytic capacity with a more pronounced reduction in male ST. On the other hand, while ST from both sexes have an efficient and higher rate of mitochondrial respiration over their respective CT, this is more pronounced in female ST. The reduced capability of male trophoblasts (ST) to shift to the more efficient mitochondrial oxidation suggests that they might not be equipped at coping with situations that require an increase in energy production. This is further evident in their reduced metabolic flexibility in using either glucose, glutamine, or fatty acids as substrates. Our results thus provide evidence for sexual dimorphism on the cellular, metabolic, and functional level in placental trophoblast. Collectively, our results fortify the notion that male placentas function at near their maximum limit, and if presented with a stress, may not be able to increase energy production and are at a higher risk of suffering from adverse pregnancy outcomes [16,46].

## 4. Materials and Methods

### 4.1. Ethical Approval of the Study

Placentae were collected from the Labor and Delivery Unit at Oregon Health and Science University into a tissue repository under a protocol approved by the Institutional Review Board with informed consent from the patients. Fetal weight was recorded. All tissues and clinical data were de-identified before being made available to the investigative team.

### 4.2. Collection of Placental Tissues

Placentae were collected and weighed immediately following Cesarean section from uncomplicated pregnancies at term in the absence of labor from patients with either a male or a female fetus and a pre-pregnancy BMI in the normal weight range (NW, BMI = 18.5–24.9, *n* = 8 male, 8 female). Exclusion criteria included overweight or obesity, multifetal gestation, gestational diabetes mellitus, preeclampsia, chronic inflammatory diseases, use of tobacco/illicit drugs, and recent bariatric surgery. Five random samples of tissue (~80 g) were collected from each placenta and placed in PBS to be transported back to the lab.

### 4.3. Primary Cell Isolation and Culture

The chorionic plate and decidua were removed from each randomly isolated placental sample, leaving only villous tissue, which was thoroughly rinsed in PBS to remove excess blood. Primary cytotrophoblast were isolated from villous tissue using a protocol adapted from Eis et al. [48] using trypsin/DNAse digestion followed by density gradient purification. Isolated cytotrophoblast cells were then frozen in freezing media (10% DMSO in FBS) and stored in liquid nitrogen until usage.

Cytotrophoblast cells were rapidly thawed in a 37 °C water bath and immediately diluted in Iscove’s modified Dulbecco’s medium (25 mM glucose, 4mM glutamine, and 1 mM pyruvate (ThermoFisher Scientific, Waltham, MA, Cat. #12440053) supplemented with 10% FBS and 1% penicillin/streptomycin (complete media) (ThermoFisher Scientific, Cat. #MT35010CV, #15140 respectively). Cells were centrifuged at 1000 × RCF for 10 min and re-suspended in fresh complete media. Trophoblast cells were plated in a 96-well Seahorse plate (100,000 cells/well) in 100µL of complete media for glycolysis and respiration measurements or plated in a 6-well plate (4 million cells/well) in 2 mL complete media for protein expression studies. The following day, additional complete media was added to each well. All studies were performed at two time points—24 hrs (labelled as cytotrophoblast/CT) and 96 hrs to allow fusion and formation of syncytiotrophoblast (ST). ST formation was confirmed by staining the cells for the trophoblast marker Cytokeratin-7.

### 4.4. Immunocytochemistry

CT cells were plated on circular coverslips at a cell density of 1.5 million cells/mL in a volume of 0.3 mL. CT (24 h) and ST (96 h) were fixed in ice-cold methanol for 10 min at −20 °C and washed three times with cold PBS. Cells were then blocked in 3% BSA diluted in PBS + 0.1% Tween 20 (PBST) for 2 hrs at room temperature. Cytokeratin-7 primary antibody (1:100) (ThermoFisher Scientific, Waltham, MA, Cat. #MA1-06315) was incubated overnight at 4 °C. Following primary antibody incubation, cells were washed three times in PBST and incubated with anti-mouse Alexa fluor 555 secondary antibodies (1:1000) (Thermofisher Scientific, Cat. #A31570) for 3 hrs at room temperature. Cells were then washed three times in PBST followed by Hoechst 33342 (1:10,000) counterstain for 30 s. Cells were washed three more times with PBST and mounted on slides using SlowFade Diamond Antifade Mountant (Thermofisher Scientific, Cat. #S36972). After allowing to set for 24 hr, cover-slips were sealed in place using clear nail polish. Images were captured using a Zeiss LSM 880 confocal microscope and processed using ImageJ Software (Bethesda, Rockville, MD, USA).

### 4.5. Metabolic Analysis and Cellular Bioenergetics Measurements

CT and ST bioenergetics were measured using Seahorse XF Analyzer (Agilent Technologies, Santa Clara, CA, USA) assays following the manufacturer’s protocol outlined briefly below. For all assays, 100,000 cells were plated per well in a 96-well Seahorse assay plate. 

#### 4.5.1. Mitochondrial Stress Test

This was used to assess mitochondrial function parameters: basal respiration, ATP production-coupled respiration, maximal respiration, spare capacity, and non-mitochondrial respiration using the Seahorse XF Cell Mito Stress Test (Agilent Technologies, Cat # 103010). One hr prior to running the mitochondrial stress test, complete media was exchanged with basal Seahorse media supplemented with glucose, glutamine, and pyruvate to match culture conditions. The cells were then allowed to equilibrate in a non-CO_2_ 37 °C incubator for 1 hr before the first rate measurement, called ‘Basal respiration rate’, and is defined as the initial oxygen consumption rate (OCR). This represents the total mitochondrial respiration rate. After measuring the baseline, 75 µL of oligomycin (ATP synthase inhibitor), FCCP (protonophore), and a combination of rotenone (NADH dehydrogenase inhibitor) and antimycin A (cytochrome c reductase inhibitor) solutions were sequentially added to each well at a 1 µM working concentration to determine the ATP coupled respiration, maximum respiration, and non-mitochondrial oxygen consumption rates, respectively. The ATP coupled response is defined the rate of oxygen consumption linked to ATP production and is calculated as the difference between the basal OCR and the OCR after oligomycin injection. Maximal respiratory rate was calculated as the difference between the OCR after uncoupled addition (FCCP) and the lowest OCR reached after oligomycin addition. Spare (reserve) capacity is calculated as the difference between OCR after FCCP and basal respiration and represents the spare metabolic potential thought to guard against stressful conditions (Figure 3A) [49]. 

#### 4.5.2. Modified Mitochondrial Stress Test

An adapted version of the mitochondrial stress test described above that was used to examine substrate impact on spare capacity by determining the rate of oxidation of a single substrate (glucose, glutamine, or long-chain fatty acids) while the other two substrate pathways are blocked. The pathway inhibitors used were 2 µM UK5099 (inhibitor of glucose oxidation, blocks action of mitochondrial pyruvate carrier (MPC), which converts glucose to pyruvate), 3 µM BPTES (inhibitor of glutamine oxidation, blocks glutaminase (GSL1), which converts glutamine to glutamate) and 4 µM Etomoxir (inhibitor of long-chain fatty acid oxidation, which blocks carnitine palmitoyltransferase 1 alpha (CPT1α). The cells were treated with either a combination of two pathway inhibitors or a combination of all three pathway inhibitors followed by the mitochondrial stress test ETC inhibitors to calculate the capacity of each pathway using the following formula.
$$\begin{array}{c} \mathrm{Substrate}\;\mathrm{impact}\;\mathrm{on}\;\mathrm{Spare}\;\mathrm{capacity}\\ \;\;\;=1-\left[\frac{\mathrm{No}\;\mathrm{OCR}\;\mathrm{inhibitor}-\mathrm{Two}\;\mathrm{OCR}\;\mathrm{inhibitors}}{\mathrm{No}\;\mathrm{OCR}\;\mathrm{inhibitor}-\mathrm{Three}\;\mathrm{OCR}\;\mathrm{inhibitors}}\right]\times 100 \end{array}$$

#### 4.5.3. Glycolysis Stress Test

This was used to assess glycolytic function parameters: glycolysis, glycolytic capacity, glycolytic reserve, and non-glycolytic acidification using the Seahorse XF Glycolysis Stress kit (Agilent Technologies, Cat # 103020). One hr prior to running the glycolysis stress test, the cell culture medium was exchanged with basal Seahorse media supplemented with glutamine (excluding glucose and pyruvate) to match culture conditions. The cells were then allowed to equilibrate in a non-CO_2_ 37 °C incubator for 1 hr before the first rate measurement called ‘Non-glycolytic acidification’ and is defined as the extracellular acidification rate (ECAR) that is not attributed to glycolysis. After measuring Non-glycolytic acidification rate, 75 µL of glucose (converted to pyruvate through glycolysis), Oligomycin (ATP synthase inhibitor), and 2-deoxyglucose-glucose (competitive inhibitor of hexokinase, the first enzyme in the glycolysis pathway) solutions were sequentially added to each well at a 10 mM glucose, 1 µM Oligomycin and 50 mM 2-deoxy-glucose working concentration to determine the rate of glycolysis under basal conditions, maximum glycolytic capacity and to confirm the initial ECAR measured is due to glycolysis, respectively. Glycolysis is defined as the glucose-induced increase in ECAR and is calculated by subtracting non-glycolytic acidification from the highest ECAR measurement following the addition of glucose. Maximum glycolytic capacity was calculated as the difference between the highest ECAR measurement during non-glycolytic acidification and the highest ECAR measurement after the addition of Oligomycin. Glycolytic reserve was calculated as the difference between ECAR after glucose and after oligomycin.

Data from all Seahorse assays were normalized to cellular DNA content measured immediately after the assay was finished. Hoescht 33342 dye (Thermofisher Scientific, Cat. #H1399) was added to each well (1:1000 final concentration) and incubated for 30 min at 37 °C with constant shaking. Fluorescence was measured using a plate reader (excitation 350 nm emission 461 nm).

### 4.6. Protein Extraction and Western Blotting 

Proteins were extracted from cultured trophoblast cells (after 24 hrs for CT fraction and after 96 hrs for ST fraction). Briefly, media was collected and frozen for ELISA analysis. To isolate protein, cells were washed in PBS followed by lysing in 100 uL RIPA buffer with added protease/phosphatase inhibitors (ThermoFisher Scientific, Cat. #89901 #A32959 respectively). Cells were then scraped, and the cell lysate transferred to a sterile 1.5 mL tube and placed on ice. Cell debris was removed by centrifuging the cell lysate at 1000 × RCF for 10 min at 4 °C and storing the supernatant at −80 °C. Total protein was quantified using the Pierce BCA Protein Assay Kit (ThermoFisher Scientific, Cat. #23225). Approximately 20 µg of protein was separated on 12% sodium dodecyl sulphate-polyacrylamide gel electrophoresis (SDS-PAGE) hand-cast gels for approximately 30 min at 30V followed by 2 hrs at 100 V and transferred for 1 hr at 100 V onto polyvinylidene difluoride (PVDF) membranes using Mini-PROTEAN tetra cell electrophoresis chamber (BioRad, Hercules, CA, Cat. # 1658004). Membranes were blocked in 5% (*w*/*v*) nonfat milk in TBS + 0.1% Tween 20 (TBST) for 1 hr and incubated with primary antibody overnight at 4 °C. On the next day, membranes were washed three times in TBST for 5 min each and incubated with HRP-conjugated secondary antibodies. Membranes were washed and incubated in Supersignal West Pico Plus ECL Substrate (ThermoFisher Scientific, Cat. #34578) for 5 min and imaged using the GBOX system (Syngene, Frederick, MD, USA). All samples were normalized to β-Actin and analyzed using Genetools software (Syngene).

The following primary antibodies were used for western blotting: Citrate Synthase (Cell Signaling Technology, Danvers, MA, USA, Cat# 14309, RRID:AB_2665545), glutamate dehydrogenase GLUD1/GLUD2 (Abcam, Cambridge, UK, Cat# ab154027), Glutaminase (Abcam, Cat# ab93434, RRID:AB_10561964), Hexokinase 2 (Cell Signaling Technology, Cat# 2867, RRID:AB_2232946), VDAC (Cell Signaling Technology, Cat# 4661, RRID:AB_10557420), PGC1α (Novus Biologicals, Littleton, CO, USA, Cat# NBP1-04676SS, RRID: AB_1522119), CPT1α (Cell Signaling Technology, Cat# 12252, RRID:AB_2797857), OXPHOS (Abcam, Cat# ab110411, RRID:AB_2756818) and β actin (Sigma-Aldrich, St. Louis, MO, USA, Cat# A2228, RRID:AB_476697). The following HRP conjugated secondary antibodies were used: goat anti-rabbit (Cell Signaling Technology, Cat# 7074, RRID:AB_2099233) and horse anti-mouse (Cell Signaling Technology, Cat# 7076, RRID:AB_330924).

### 4.7. Enzyme Linked Immunosorbent (ELISA) Assay

The levels of human chorionic gonadotropin (hCG) hormone were measured in media collected from CT and ST cells using an ELISA based assay (R&D Systems, Minneapolis, MN, Cat. #DY9034-05) following manufacturer instructions. Data were then normalized to cellular protein measured using the Pierce BCA Protein Assay Kit (ThermoFisher Scientific, Cat. #23225). 

### 4.8. Citrate Synthase Activity

Citrate synthase activity was measured using the citrate synthase activity kit (Millipore Sigma, St. Louis, MO, USA, Cat. #MAK193) following manufacturer instructions. Briefly, 2 × 10^6^ cells/well were plated in 12-well tissue-culture plates. At 24 hrs and 96 hrs cells were lysed using 90 µL ice cold CS Assay Buffer. The total protein in the lysate was determined using Pierce BCA Protein Assay Kit (ThermoFisher Scientific, Cat. #23225) and all samples were adjusted to 40 μg of protein/50 μL using the CS assay buffer. 50 μL of the lysate was transferred to a 96-well reaction plate along with the standards supplied in the kit. 50 μL Reaction buffer was added to each well and an initial absorbance was measured at 412 nm. The plate was incubated at 25 °C for a total of 10 min before the final measurement was taken. The CS activity was calculated as S_a_/(Reaction Time) × S_v_; where Sa = Amount of GSH (nmole) generated in unknown sample well between T_initial_ and T_final_ from standard curve, Reaction Time = T_final_ − T_initial_ (minutes) and Sv = sample volume (mL) added to well. CS activity is reported as pmole/min/μL = microunit/μL. 

### 4.9. Quantitation of Mitochondrial Content

To quantitate mitochondrial number CT cells were plated in a 96-well tissue-culture dish at a cell density of 1 million cells/mL in a volume of 0.1 mL/well for 24 hrs (CT) or 96 hrs (ST). Cells were then incubated with 200 nM MitoTracker^TM^ Deep Red (Thermo Fisher Scientific, Cat. #M22426) diluted in HBSS for 30 min at 37 °C. Cells were washed three times in HBSS and MitoTracker^TM^ fluorescence (excitation 644 nm/emission 665 nm). MitoTracker^TM^ Deep Red specifically stains the mitochondria, and the OD data was normalized to DNA content measured using Quant-it Pico Green dsDNA Reagent (Thermo Fisher Scientific, Cat. #P7581). 

### 4.10. Statistical Analysis

Data are reported as box-and-whisker plots (min to max with mean) with individual data points. Data separated by fetal sex are reported as individual symbols and lines. Statistical significance between groups was calculated using the Friedman test, Wilcoxon test or paired t-test where appropriate. * *p* < 0.05, ** *p* < 0.01, and *** *p* < 0.001 are reported as statistically significant. Graphpad Prism was used to perform all statistical analyses and to generate all graphs.

## 5. Conclusions

The current study outlines fundamental differences between CT and ST energy metabolism, their responses to stressful conditions and how these are influenced by fetal sex. The study justifies further research into how exposure to in utero adverse conditions, like diabetes and obesity, might affect placental function and emphasizes the need for understanding these in the context of sexual dimorphism.

## Figures and Tables

**Figure 1 ijms-22-10875-f001:**
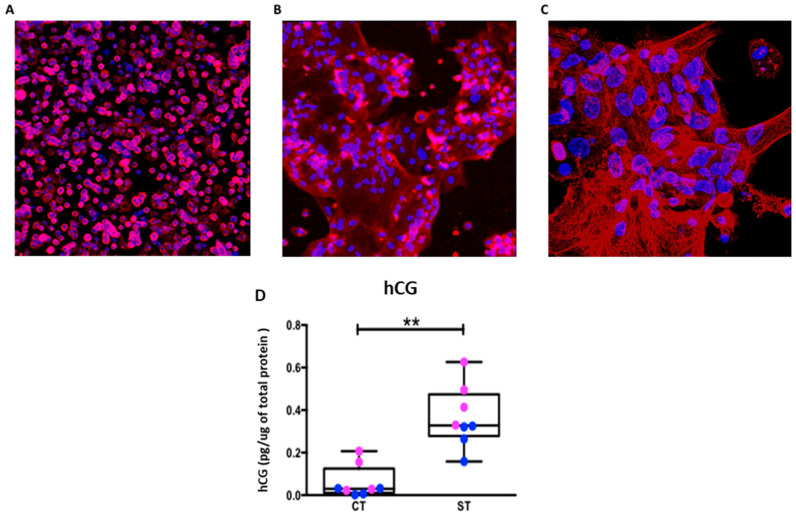
Identification of trophoblast cells and their syncytialization. (**A**) Cytotrophoblast at 24 h (20×), (**B**) Syncytiotrophoblast at 96 hrs (20×), and (**C**) Syncytiotrophoblast (63×) stained with cytokeratin 7 (red) and counterstained with Hoechst 33,342 for nuclei (blue). (**D**) Human Chorionic Gonadotropin (hCG) production pg of hormone per μg of cell protein. Data presented as minimum, maximum, median, 25th and 75th quartiles boxes, and whisker plots, *n* = 8, male = blue, female = pink. ** *p* < 0.01, (Wilcoxon test CT vs. ST).

**Figure 2 ijms-22-10875-f002:**
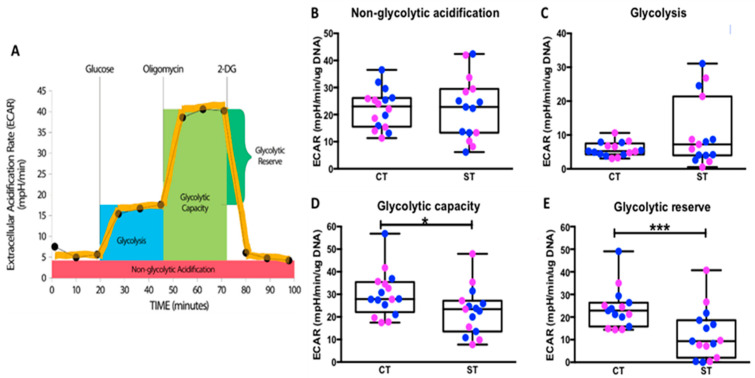
Glycolytic function of CT vs. ST analyzed using the glycolysis stress test. (**A**) Graphical representation of the glycolysis stress test, (**B**) non-glycolytic acidification, (**C**) glycolysis, (**D**) glycolytic capacity, and (**E**) glycolytic reserve. Male (blue, *n* = 8) and female (pink, *n* = 8) groups combined. Data presented as minimum, maximum, median, 25th and 75th quartiles boxes, and whisker plots. * *p* < 0.05, *** *p* < 0.001 (Wilcoxon signed-rank test). 2-DG: 2-deoxy-glucose, ECAR: extracellular acidification rate.

**Figure 3 ijms-22-10875-f003:**
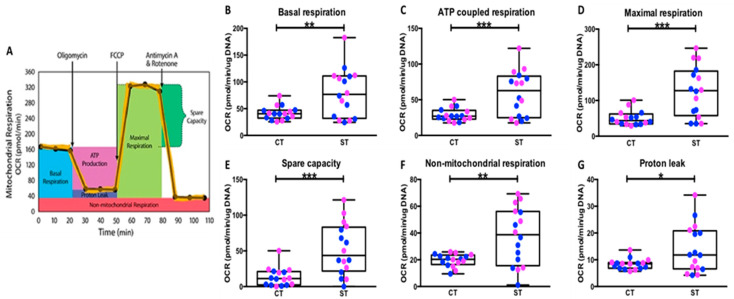
Mitochondrial respiration of CT vs. ST analyzed using the mitochondrial stress test. (**A**) Graphical representation of the mitochondrial stress test, (**B**) basal respiration, (**C**) ATP-coupled respiration, (**D**) maximal respiration, (**E**) spare capacity, (**F**) non-mitochondrial respiration, and (**G**) proton leak. Male (blue, *n* = 8) and female (pink, *n* = 8) groups combined. Data presented as minimum, maximum, median, 25th and 75th quartiles boxes, and whisker plots. * *p* < 0.05, ** *p* < 0.01, *** *p* < 0.001, and Wilcoxon signed-rank test. FCCP: Trifluoromethoxy carbonylcyanide phenylhydrazone.

**Figure 4 ijms-22-10875-f004:**
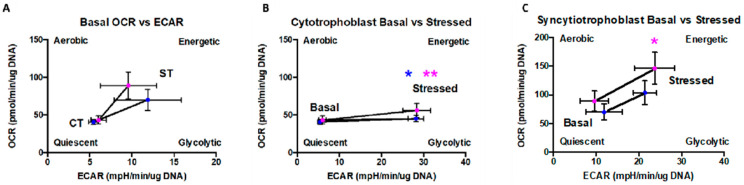
Comparison of metabolic phenotypes of CT and ST. (**A**) Metabolic shift of CT to ST and metabolic potential of (**B**) CT and (**C**) ST. Data presented as mean +/− SEM. Male (blue, *n* = 8) and female (pink, *n* = 8) groups separated. * *p* < 0.05, ** *p* < 0.01, Wilcoxon signed-rank test.

**Figure 5 ijms-22-10875-f005:**
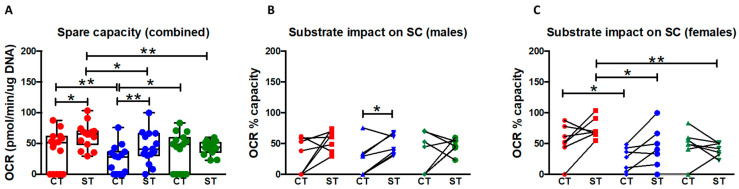
Impact of specific substrates on spare capacity of CT vs. ST. (**A**) Impact of glucose (red), glutamine (blue), and long chain fatty acid (green) on spare capacity for each fuel source with fetal sex groups combined (*n* = 16). Combined data presented as minimum, maximum, median, 25th and 75th quartiles boxes, and whisker plots. * *p* < 0.05, ** *p* < 0.01, (Friedman test when comparing substrates or Wilcoxon test when comparing CT vs. ST). (**B**) Substrate impact on spare capacity in males (*n* = 8), and (**C**) females (*n* = 8). Glucose (red), glutamine (blue), and long-chain fatty acid (green). Data plotted as individual values of paired CT and ST from the same sample. * *p* < 0.05, (Friedman test when comparing substrates or the Wilcoxon test when comparing CT vs. ST).

**Figure 6 ijms-22-10875-f006:**
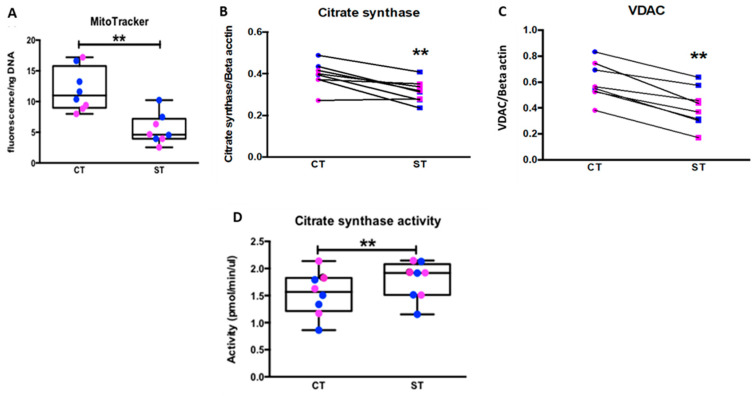
Mitochondrial content and activity measurements in cyto- and syncytiotrophoblast. (**A**) MitoTracker^TM^, (**B**) citrate synthase protein, and (**C**) VDAC protein levels. (**D**) Citrate synthase activity (in picomole/min/μL of substrate). Male (blue, *n* = 4) and female (pink, *n* = 4). A, D: Data presented as minimum, maximum, median, 25th and 75th quartiles boxes, and whisker plots. (**B**,**C**): Data plotted as individual values of paired CT and ST from the same sample Male (blue, *n* = 4) and female (pink, *n* = 4) fetal sex groups combined. ** *p* < 0.01, (Wilcoxon test, CT vs. ST).

**Figure 7 ijms-22-10875-f007:**
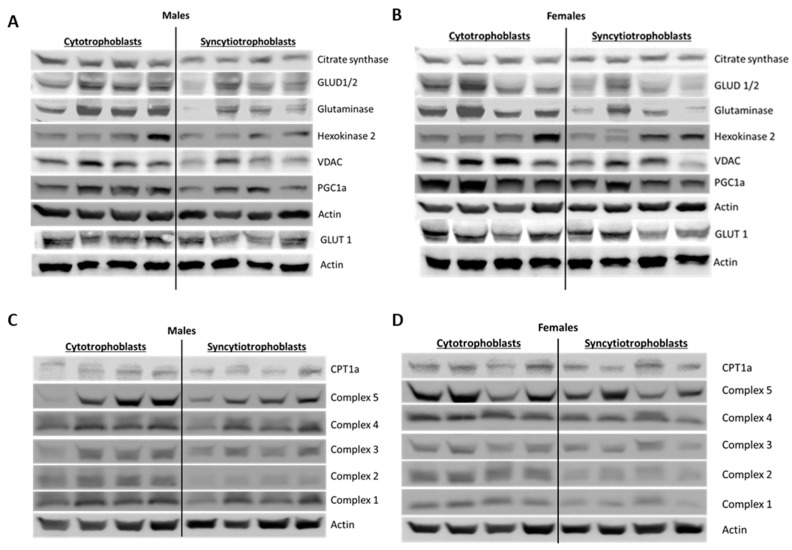
Effect of trophoblast differentiation on specific mitochondrial protein expression. Representative western blots (**A**–**D**) and quantification (**E**–**O**) of cellular proteins in CT vs. ST. Data plotted as individual values of paired CT and ST from the same sample Male (blue, *n* = 4) and female (pink, *n* = 4) fetal sex groups combined. * *p* < 0.05, ** *p* < 0.01, (Wilcoxon test, CT vs. ST).

**Table 1 ijms-22-10875-t001:** Clinical characteristics of study participants.

Fetal Sex	Pre-Pregnancy BMI (kg/m^2^)	Maternal Age (yrs)	Gestational Age (wks)	Fetal Weight (Grams)	Placental Weight (Grams)	Fetal/ Placental Ratio	Gestational Weight Gain (kg)	Ethnicity (Hispanic, Non-Hispanic)
Males *n* = 8	22.9 ± 1.6	35.9 ± 6.7	39.0 ± 0.5	3612 ± 257	508 ± 87.6	7.2 ± 1.1	15.0 ± 3.7	0, 8
Females *n* = 8	22.3 ± 1.5	32.1 ± 4.5	38.6 ± 1.0	3208 * ± 400	518 ± 71.9	6.2 * ± 0.6	15.1 ± 4.2	1, 7

Data presented as mean ± SD. Significant differences between male and female groups were determined using the student’s *t* test. * *p* < 0.05 male vs. female.

**Table 2 ijms-22-10875-t002:** List of mitochondrial metabolism proteins assessed by western blotting grouped in 3 sub-groups (capitalized).

ELECTRON TRANSPORT CHAIN COMPLEXES
NADH reductase (Complex I)
Succinate dehydrogenase (Complex II)
Cytochrome C reductase (Complex III)
Cytochrome C oxidase (Complex II)
ATP synthase (Complex V)
METABOLITE PROCESSING ENZYMES
Glutamate dehydrogenase, Mitochondrial (GLUD 1/2)
Carnitine palmitoyl transferase one alpha (CPT1α)
Hexokinase 2
Glutaminase
Glucose Transporter Type 1(GLUT1)
MITOCHONDRIAL BIOGENESIS
Peroxisome proliferator-activated receptor gamma coactivator 1-alpha (PGC1α)

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
