# Peer review of "Differences in Glycolysis and Mitochondrial Respiration between Cytotrophoblast and Syncytiotrophoblast In-Vitro: Evidence for Sexual Dimorphism"

_ijms, 2021, doi:10.3390/ijms221910875_

Round 1
Reviewer 1 Report
Paper revision:
The authors provide evidence of physiological differences between partially differentiated cytotrophoblasts and terminally differentiated syncytiotrophoblast from term human placenta as regard to bioenergetics and glucose metabolism. They also prove sexual dimorphism in some mitochondrial activity and protein levels. The study is very important in the topic of placenta physiology and function, and tends to explain some of the previous publications from sex-dependent differences in placental gene expression. The article also brings contradictory evidence for the mitochondrial function when cytotrophoblasts are undergoing differentiation, which is underscore by the use of a more physiologic model (purified CT) in this article as respect to previous publications (that use purified mitochondria). Thus, this paper is very important to make some light in the field. The introduction include all relevant references and brings sufficient background for the study. The experimental approach are appropriate, sufficiently described and results are carefully interpreted and discussed.
Minor revision:
Point 1 – Can authors explain the difference in the ratio of hispanic vs non-hispanic between placentas from males and females groups (0.8 versus 1.7). Can authors bring evidence to exclude the ethnicity influence in gene expression and thus the difference in profile between male vs female in this study? Can authors address this in the discussion section and if available bring references of absence of differences between ethnicity as regard to placental gene expression, including mitochondria.
Point 2 – All along the paper figures authors address potential differences between males and females groups and along trophoblast terminal differentiation (CT versus ST). However, this is not the case in Figure 1 were males versus females trophoblasts are not separated as regard to their morphological (CT vs ST) and functional (hCG production) differentiation. Is there a difference of the ability of female cytotrophoblasts to differentiate when compared to male cytotrophoblasts that could further explain the differences in mitochondria protein expression? Can authors prove/disprove this by hormone secretion and imaging of ST formation and add a comment in the discussion section?
Point 3 – For the differentiation ability of CT in Fig 1 A authors choose to use Cytokeratin 7 staining which is specific for trophoblasts, but this is not the best marker to distinguish between CT and ST. When responding to Point 2 can authors stain membrane proteins (E-Cadherin, Desmoplakin…) in order to better identify ST (multinucleated cells delimitated by membrane staining). They can also do “fusion index” count (% of number of syncytial nuclei/total nuclei), with is the most appropriate and commonly admitted for CT differentiation in ST.
Point 4 – Although is appreciable that the authors separate the results for better visualisation, can authors also represent the results of acidification and OCR directly in the graphics of Fig 2A and Fig 3A as regard to CT vs ST (at least as supplementary information if not in the main figure).
Point 5 – It could be interesting and informative to see the difference between CT ans ST from Mitotracker images and not only results in a graph (Fig 6A). Can authors put these images in supplemental figures?
Optional point (not mandatory) – Do authors have data from RIESKE sub-unit of Complex III, NDUFS3 s/u of Complex II and/or mitochondria Aconitase at protein level (Western Blots Fig 5). These are Fe-S cluster proteins, highly sensitive to environmental stress (oxidative stress…), that could be interesting to check in the approach of CT vs ST and/or sex-differences in responses to stress as underlined by the authors.
Other minor points:
- Please correct some typographic errors (overuse of spaces)
- Please explain the abbreviations when they appears for the first time in the text: BMI (line 92), 2-DG (Fig 2 legend), FCCP (Fig 3 legend), ECAR (Fig 1 legend)
- I do not understand at what time ST were stopped and analysed: 72h (lines 106, 119…) or 96h (line 333)?
Author Response
We thank the reviewer for a critical consideration of our work and for providing insightful comments. We agree with the reviewer on most of the points and made the necessary edits in manuscript. A detailed response to their comments is as follows:
The authors provide evidence of physiological differences between partially differentiated cytotrophoblasts and terminally differentiated syncytiotrophoblast from term human placenta as regard to bioenergetics and glucose metabolism. They also prove sexual dimorphism in some mitochondrial activity and protein levels. The study is very important in the topic of placenta physiology and function, and tends to explain some of the previous publications from sex-dependent differences in placental gene expression. The article also brings contradictory evidence for the mitochondrial function when cytotrophoblasts are undergoing differentiation, which is underscore by the use of a more physiologic model (purified CT) in this article as respect to previous publications (that use purified mitochondria). Thus, this paper is very important to make some light in the field. The introduction include all relevant references and brings sufficient background for the study. The experimental approach are appropriate, sufficiently described and results are carefully interpreted and discussed.
Minor revision:
Point 1 – Can authors explain the difference in the ratio of hispanic vs non-hispanic between placentas from males and females groups (0.8 versus 1.7). Can authors bring evidence to exclude the ethnicity influence in gene expression and thus the difference in profile between male vs female in this study? Can authors address this in the discussion section and if available bring references of absence of differences between ethnicity as regard to placental gene expression, including mitochondria.
Response: We apologize to the reviewer for how the data is presented in table 1 which may have led to some confusion. The data is not a ratio but shows the number of Hispanic and non-Hispanic patients: all 8 males being non-hispanic, and 7 out of the 8 females being non-hispanic. We have made this clearer in the table. We agree with the reviewer that effect of ethnicity on gene expression may be a crucial aspect; however, we do not have enough patients to comment on this aspect.
Point 2 – All along the paper figures authors address potential differences between males and females groups and along trophoblast terminal differentiation (CT versus ST). However, this is not the case in Figure 1 were males versus females trophoblasts are not separated as regard to their morphological (CT vs ST) and functional (hCG production) differentiation. Is there a difference of the ability of female cytotrophoblast to differentiate when compared to male cytotrophoblast that could further explain the differences in mitochondria protein expression? Can authors prove/disprove this by hormone secretion and imaging of ST formation and add a comment in the discussion section?
Response: The reviewer raises an interesting point. We provide the fetal sex-based differences in hCG production in supplementary figure 1. As previously reported in literature [1-3], we also observe higher production of hCG in female trophoblasts. We include the data on hCG as evidence for biochemical differentiation (and images in Fig 1A for morphological differentiation) to illustrate that cytotrophoblast (CT) differentiate to syncytiotrophoblast (ST) in our culture conditions and confirm our trophoblast cell types used for the rest of the study. Correlation between higher hCG production in female trophoblast and their rate of differentiation has not been indicated in literature and would be interesting to explore. We think it would have to be a detailed time course experiment utilizing high resolution imaging to determine differences in the rate of morphologic differentiation Corelating these differences (if any) to mitochondrial protein expression would then be an independent study and is thus beyond the scope of this manuscript.
Point 3 – For the differentiation ability of CT in Fig 1 A authors choose to use Cytokeratin 7 staining which is specific for trophoblasts, but this is not the best marker to distinguish between CT and ST. When responding to Point 2 can authors stain membrane proteins (E-Cadherin, Desmoplakin) in order to better identify ST (multinucleated cells delimitated by membrane staining). They can also do “fusion index” count (% of number of syncytial nuclei/total nuclei), with is the most appropriate and commonly admitted for CT differentiation in ST.
Response: It is well established and widely accepted in literature that isolated term CT spontaneously fuse to from ST in culture conditions [4]. We provide Fig 1A as a reiteration that in our culture system, CT do fuse to from ST in addition to secretion of hCG which is also an accepted marker for syncytialization [5, 6]. However, as per the reviewer’s suggestion and to further strengthen our data, we now provide E-Cadherin staining as Supplemental Figure 1B (i, ii).
Point 4 – Although is appreciable that the authors separate the results for better visualization, can authors also represent the results of acidification and OCR directly in the graphics of Fig 2A and Fig 3A as regard to CT vs ST (at least as supplementary information if not in the main figure).
Response: We presume the reviewer is asking for a graphic of the OCR and ECAR tracing for matched CT and ST from the same placenta. Including traces from all 8 male and female placentas on the same graph would be messy and confusing. However, to provide the reader with a comprehensive overview, we now provide a representative tracing of one male and female placenta as supplementary figure 2 (E-H) and 3 (G-J).
Point 5 – It could be interesting and informative to see the difference between CT and ST from Mitotracker images and not only results in a graph (Fig 6A). Can authors put these images in supplemental figures?
Response: The mitochondrial mass was determined by measuring the absorbance of MitoTrackerTM using a fluorescence microplate reader (as per manufacturer’s protocol). We therefore do not have MitoTrackerTM images from these cells. But we do agree with the reviewer that it would be interesting to see the differences between CT & ST mitochondrial structure and are currently pursuing electron microscopy imaging studies for the same.
Optional point (not mandatory) – Do authors have data from RIESKE sub-unit of Complex III, NDUFS3 s/u of Complex II and/or mitochondria Aconitase at protein level (Western Blots Fig 5). These are Fe-S cluster proteins, highly sensitive to environmental stress (oxidative stress…), that could be interesting to check in the approach of CT vs ST and/or sex-differences in responses to stress as underlined by the authors.
Response: We currently do not have this data but agree it would be interesting to have it.
Other minor points:
- Please correct some typographic errors (overuse of spaces)
- Please explain the abbreviations when they appears for the first time in the text: BMI (line 92), 2-DG (Fig 2 legend), FCCP (Fig 3 legend), ECAR (Fig 1 legend)
- I do not understand at what time ST were stopped and analyzed: 72h (lines 106, 119…) or 96h (line 333)?
Response: We have corrected the typographic errors and the abbreviations are now explained the first time they were used as suggested.
We apologize for the confusion regarding the timing. All assays were performed at 96 hours and we have corrected the errors in lines 106, 119.
REFERENCES:
- Adibi, J.J., et al., Fetal sex differences in human chorionic gonadotropin fluctuate by maternal race, age, weight and by gestational age. J Dev Orig Health Dis, 2015. 6(6): p. 493-500.
- Yaron, Y., et al., Maternal serum HCG is higher in the presence of a female fetus as early as week 3 post-fertilization. Hum Reprod, 2002. 17(2): p. 485-9.
- de Graaf, I.M., et al., Co-variables in first trimester maternal serum screening. Prenat Diagn, 2000. 20(3): p. 186-9.
- Li, L. and D.J. Schust, Isolation, purification and in vitro differentiation of cytotrophoblast cells from human term placenta. Reproductive Biology and Endocrinology, 2015. 13(1): p. 71.
- Ho, H.H., et al., The relationship between trophoblast differentiation and the production of bioactive hCG. Early Pregnancy, 1997. 3(4): p. 291-300.
- Hoshina, M., et al., The Role of Trophoblast Differentiation in the Control of the hCG and hPL Genes, in Human Trophoblast Neoplasms, R.A. Pattillo and R.O. Hussa, Editors. 1984, Springer US: Boston, MA. p. 299-312.
Reviewer 2 Report
Differences in glycolysis and mitochondrial respiration between cytotrophoblast and syncytiotrophoblast in-vitro: evidence for sexual dimorphism by Matt Bucher, Leena Kadam, Kylia Ahuna and Leslie Myatt
This is an interesting paper. My major concerns are the novelty and conclusions of the paper.
The authors confirm/dispute previously published data on different metabolic capacity of CTB vs STB published by Kolahi, K. S. et al. (Cytotrophoblast, Not Syncytiotrophoblast, Dominates Glycolysis and Oxidative Phosphorylation in Human Term Placenta. Sci. Rep. 7, 42941; doi: 10.1038/srep42941) but fail to properly discuss it in their manuscript. Kolagi et al used the same methodological approach, the same analysis, and published either similar or contradictory data – I am alarmed that the authors do not mention these facts in their study!?! I am sure they are aware of this paper since they mention it – very briefly – in the introduction but it must be discussed properly in the Discussion part of the manuscript and explicitly clarify to the reader what is new in this manuscript.
For example:
Lines 320-321 the authors must admit that this has been previously shown by others.
Lines 351-370 – the authors found that mitochondrial respiration is higher in STB and say that their data contrasts those by Fisher et al. but explain it by the fact that Fisher used isolated mitochondria whilst they worked with isolated cells. However, Kolahi et al reports actually the same findings as Fisher (higher oxygen consumption, mitochondrial respiration and ATP levels in CTB) working with the same cells as Bucher et al. So, the point is not valid and the authors must elaborate the whole paragraph and discuss also the findings of Kolahi et al.
In general, the Discussion part has to be revised radically to meet scientific standards!
Regarding novelty, the really new point I see in the manuscript is the sexual dimorphism – which, on the other hand is an interesting and important finding.
However, in the Abstract, the authors claim that they “…have shown that with adverse in-utero conditions of diabetes and obesity trophoblast respiration is sexually dimorphic”. This is a considerable overstatement as the authors did not provide experimental evidence to that in the current manuscript. These exaggerations must be removed from the text (Abstract and Conclusion) or supported by clear data.
Minor points:
Materials and Method section:
Line 466: symbol µ is missing
Line 556: symbol µ is missing
Line 591: symbol µ is missing
….maybe some more, I believe these could have been introduced during PDF conversion – anyway, please double check during revision.
Results:
Legend of figure 1: *p<0.05 (it is not in the figure, significance with 1 star) – can be removed
Legend of figure 2: **p<0.01 (It is not in the figure, significance with 2 stars) – can be removed
Legend figure 6: *p<0.05 (it is not in the figure, significance with 1 star) – can be removed
Figure 5 (B and C) would benefit from making the same scale on Y axis to be able to see the differences more clearly.
Author Response
We thank the reviewer for their time and a critical consideration of our work. We have made the necessary edits in manuscript and a detailed response to their comments is as follows:
This is an interesting paper. My major concerns are the novelty and conclusions of the paper.
The authors confirm/dispute previously published data on different metabolic capacity of CTB vs STB published by Kolahi, K. S. et al. (Cytotrophoblast, Not Syncytiotrophoblast, Dominates Glycolysis and Oxidative Phosphorylation in Human Term Placenta. Sci. Rep. 7, 42941; doi: 10.1038/srep42941) but fail to properly discuss it in their manuscript. Kolagi et al used the same methodological approach, the same analysis, and published either similar or contradictory data – I am alarmed that the authors do not mention these facts in their study!?! I am sure they are aware of this paper since they mention it – very briefly – in the introduction but it must be discussed properly in the Discussion part of the manuscript and explicitly clarify to the reader what is new in this manuscript.
For example:
Lines 320-321 the authors must admit that this has been previously shown by others.
Lines 351-370 – the authors found that mitochondrial respiration is higher in STB and say that their data contrasts those by Fisher et al. but explain it by the fact that Fisher used isolated mitochondria whilst they worked with isolated cells. However, Kolahi et al reports actually the same findings as Fisher (higher oxygen consumption, mitochondrial respiration and ATP levels in CTB) working with the same cells as Bucher et al. So, the point is not valid and the authors must elaborate the whole paragraph and discuss also the findings of Kolahi et al.
In general, the Discussion part has to be revised radically to meet scientific standards!
Regarding novelty, the really new point I see in the manuscript is the sexual dimorphism – which, on the other hand is an interesting and important finding.
Response: Careful reading of the study published by Kolahi, K. S. et al revelas that indeed it uses a similar model and assay but with very different experimental conditions and analysis. The Kolahi study focuses on fatty acid metabolism and the energy cost of fatty acid esterification. Their experiments were therefore performed in media with addition of high levels of fatty acids C16:0 (Palmitate) and C18:1 (Oleate) and thus do not reflect the basal metabolic status of CT and ST which was the aim of our study. The study also reports higher baseline glycolysis rate (no statistical difference in our study), higher maximal capacity (like ours) and no differences in glycolytic reserve (statistically higher in CT in our study). In Seahorse assay, glycolysis is measured by ECAR i.e. lactate production from glycolysis. In the Kolahi, K. S. et al study however, these analyses were performed in media supplemented with pyruvate – the intermediate before lactate in glycolysis. The presence of pyruvate in the media thus confounds their baseline data and their conclusions. Additionally, as the reviewer acknowledges, they did not study sexual dimorphism as we have here. We appreciate that the reviewer finds this interesting and important. We apologize for initially “alarming” the reviewer, and now address their concerns in lines 346-355 and 381-384.
However, in the Abstract, the authors claim that they “…have shown that with adverse in-utero conditions of diabetes and obesity trophoblast respiration is sexually dimorphic”. This is a considerable overstatement as the authors did not provide experimental evidence to that in the current manuscript. These exaggerations must be removed from the text (Abstract and Conclusion) or supported by clear data.
Response: Our group, in a previously published study Wang et al 2019, [1] has shown that diabetes and obesity affect trophoblast respiration in a sexually dimorphic manner and this is referenced in the discussion (lines 432-436). The sentence now reads as “We have previously shown that with adverse in-utero conditions of diabetes and obesity trophoblast respiration is sexually dimorphic” We edited the abstract to clarify this.
Minor points:
Materials and Method section:
Line 466: symbol µ is missing
Line 556: symbol µ is missing
Line 591: symbol µ is missing
….maybe some more, I believe these could have been introduced during PDF conversion – anyway, please double check during revision.
Response: The symbols were indeed introduced during the PDF conversion, and we have the changed these back in the new version. On the author side, we do not get a pdf converted version, so we cannot determine if they are replaced again. But we will keep in mind to check for these in the proof copy.
Results:
Legend of figure 1: *p<0.05 (it is not in the figure, significance with 1 star) – can be removed
Legend of figure 2: **p<0.01 (It is not in the figure, significance with 2 stars) – can be removed
Legend figure 6: *p<0.05 (it is not in the figure, significance with 1 star) – can be removed
Figure 5 (B and C) would benefit from making the same scale on Y axis to be able to see the differences more clearly.
Response: We again thank the reviewer for above comments and have made the suggested edits in the legends. We also adjusted the Y axis for Fig 5B & C.
RFERENCES:
- Wang, Y., M. Bucher, and L. Myatt, Use of Glucose, Glutamine and Fatty Acids for Trophoblast Respiration in Lean, Obese and Gestational Diabetic Women. J Clin Endocrinol Metab, 2019. 104(9): p. 4178-87.
Round 2
Reviewer 2 Report
The authors have addressed all concerns of mine. The only remaining minor point is to add reference number [5] in places where Kolahi´s paper is discussed (lines 347 and 379).